# Electrical Properties of Poly(Monomethyl Itaconate)/Few-Layer Functionalized Graphene Oxide/Lithium Ion Nanocomposites

**DOI:** 10.3390/polym12112673

**Published:** 2020-11-12

**Authors:** Quimberly Cuenca-Bracamonte, Mehrdad Yazdani-Pedram, Marianella Hernández Santana, Héctor Aguilar-Bolados

**Affiliations:** 1Facultad de Ciencias Químicas y Farmacéuticas, Universidad de Chile, Olivos 1007, Santiago 8380544, Chile; cuencaq@postqyf.uchile.cl; 2Instituto de Ciencia y Tecnología de Polímeros, ICTP-CSIC, Juan de la Cierva, 3 28006 Madrid, Spain; marherna@ictp.csic.es

**Keywords:** sustainable polymer, Li^+^-graphene-based nanocomposites, electrical properties

## Abstract

Poly(monomethyl itaconate) is outstanding because it is a glassy and dielectric polymer obtained from sustainable feedstock. Consequently, the study of the properties of its nanocomposites has gained importance. Herein, the electrical properties of nanocomposites based on poly(monomethyl itaconate) and functionalized few-layer graphene oxide (FGO) in the presence and absence of lithium ions (Li^+^) are studied. Not only did the electrical conductivities of the nanocomposites present values as high as 10^−5^ Scm^−1^, but also the dielectric permittivity of nanocomposites with (FGO) content lower than the percolation threshold was twice that of the pristine polymer, without presenting a drastic increase of the loss tangent. By contrast, nanocomposites containing Li^+^ ions presented significant increases of the permittivity with concomitant increases of the loss tangent. Moreover, it was determined that the presence of Li^+^ ions influenced the charge transport in the composites because of its ionic nature.

## 1. Introduction

Polymers obtained from sustainable sources have gradually captured the attention of the scientific community [1]. That is mainly attributable to the environmental issues associated with the production of petroleum-based polymers, the pollution associated with the improper management of their disposal and the lack of recycling initiatives [2]. In this regard, an interesting sustainable feedstock for the production of polymers is itaconic acid, which stands out because it is produced by fungal fermentation of sugars [3]. Furthermore, the interest in the polymerization of itaconic acid and its derivative has caught the attention of researchers since the 50s [4,5], especially in regard to a sustainable approach [6]. The presence of a vinyl group in this dicarboxylic acid compound allows its use as monomer in radical chain-growth polymerization. However, the proximity of carboxylic acid groups to the vinyl group limits the radical polymerization yield due to the electron withdrawing nature of the carboxylic acids. The mono-esterification of the itaconic acid by using an alcohol such as methanol is a simple strategy for solving this issue because the effect of electron withdrawing from the vinyl group will be suppressed [7]. In spite of a number of monoesters, diesters and mixed-esters of itaconic acid have been synthesized and polymerized by radical polymerization [8]. The simplest and most inexpensive polymer based on itaconic acid derivative is poly(monomethyl itaconate). Overall, itaconic acid-based polymers are glassy polymers and considered as dielectric polymers [9]. This fact has awoken the interest in investigating their use for the development of organic-based electrodes and capacitors [10]. Some glassy polymers present high glass transition temperatures. For instance, it has been reported that the glass transition temperature of poly(monomethyl itaconate) is 435 K (ca. 172 °C) [6]. This indicates that the glass transition temperature of this polymer is higher than the degradation temperatures of the pendant groups which start above 145 °C [11].

The majority of glassy polymers have discrete electrical properties, such as very low electrical conductivity (σ ca. 10^−15^ S cm^−1^), so different strategies have been explored in order to impart more attractive electrical properties [12]. In this regard, the advent of graphenic materials has provided interesting alternatives for modifying a wide range of polymers [13]. This stems from the unique properties of graphene, such as high electrical and thermal conductivities, low joule effect and excellent mechanical properties [14]. As is known, graphene corresponds to a hexagonal long-range π-conjugated system of monoatomic thickness, and the graphenic materials have similar structures and properties to graphene containing functional groups that favor the interaction with the polymer matrix [15,16]. The graphenic materials impart electrical properties, i.e., electrical conductivity, because the nanomaterials dispersed through the polymer matrix form an electrical percolation network, favoring the charge transport by electron hopping [17]. Besides, the presence of carbon-based materials of this type provides inner polarization at mesoscopic scale, resulting in increases of dielectric properties [18]. Moreover, the influence on the charge transport of ions, such as Li^+^, provides an interesting topic to study, since the presence of the Li^+^ ions can impart additional polarization [19].

Herein, a basic study is reported in order to explore the nature of the electrical conductivity in polymer composites based on a graphenic material and poly(monomethyl itaconate) in the presence and absence of Li^+^ ions. The graphenic material corresponds to a reduced graphene oxide functionalized with poly(monomethyl itaconate) (FGO) developed by our group [20]. The use of this FGO as filler in poly(monomethyl itaconate) matrix is addressed to enhance the interaction between the surface of the graphenic material and the polymer matrix. This study provides interesting background related to the composites based on glassy sustainable polymers useful for the development of glassy polymer-based electrodes and capacitors.

## 2. Materials and Methods

### 2.1. Materials

Monomethyl itaconate was obtained from itaconic acid (≥99.0%) using a previously reported method [21]. Benzoyl peroxide (75%) and lithium hydroxide (98%) were supplied by Sigma Aldrich (St. Louis, MO, USA). FGO was synthetized using a methodology developed by our group [20]. This FGO consisted of a few-layer reduced graphene oxide containing 49% of poly(monomethyl itaconate) covalently bound to its surface. The number of stacked graphene layers was between 3 and 6.

### 2.2. Synthesis of Poly(Monomethyl Itaconate) and Poly(Monomethyl Itaconate) Li^+^

In total, 14.2 g of monomethyl itaconate was added to a round bottom reactor flask. Then, the reactor was sealed with a septum and immersed in a heating bath at 74 °C. Once the monomer was melted, 0.1221 g of benzoyl peroxide was added and the reactor was purged using nitrogen gas. The bath temperature was then increased to 80 °C and the reaction was left for 12 h. Once the reaction was concluded (time expired), the resulting solid was dissolved in methanol and reprecipitated in dimethyl ether.

In order to prepare poly(monomethyl itaconate) Li^+^ (PMMI-Li^+^), 1 g of poly(monomethyl itaconate) (PMMI) was added to 20 mL of distilled water and was left under stirring for 1 h to achieve complete dissolution of the polymer. The resulting pH of the polymer solution was of 3. After, 187 mg of lithium hydroxide was added and the solution was stirred for 1 h. The pH of the resulting solution was 12. Then, this solution was poured into a Petri dish and left to dry in a ventilated oven at 40 °C for 12 h. Finally, the solid was dried at 40 °C under vacuum until reaching constant weight.

### 2.3. Preparation of Nanocomposites

Nanocomposites were prepared by dissolving 1 g of PMMI or PMMI-Li^+^ in a determined volume of distilled water. Then, determined aliquots of FGO suspension were added in order to achieve contents of 0.5%, 1%, 3%, 5% and 10% of FGO with respect to the weight of the polymer. Afterwards, the resulting dispersions were sonicated for 15 min using a Q700 sonicator, Qsonica (Newtown, CT, USA). Then the resulting dispersions were each poured into a Petri dish and dried in a ventilated oven at 40 °C for 12 h. Finally, the solid was dried in a vacuum oven at 40 °C and 120 Torr until reaching constant weight.

### 2.4. Characterization

FTIR spectra were recorded using a Thermo Scientific Nicolet IS5 spectrophotometer (Waltham, MA, USA) with the attenuated total reflectance technique (ATR). The Raman spectra were registered using a Renishaw Invia Raman microscope (Wottom-under-Edge, United Kingdom) equipped with a 785 nm wavelength laser and 0.02 cm^−1^ resolution for characterization of PMMI and PMMI-Li^+^, and for recording the Raman spectrum of FGO a 514 nm laser was used. Besides, the X-ray diffraction of FGO was recorded using a Bruker D8 Advance diffractometer (Billerica, MA, USA). The radiation frequency used was the Kα line from Cu (1.5406 Å) with a power supply of 40 kV and 40 mA. The morphologies of the samples were obtained by scanning electron microscopy (SEM) using JSM-IT300LV microscopy, Jeol (Tokyo, Japan). In order to improve the quality of the images, the samples were coated with an ultra-thin gold/palladium (Au/Pd) layer. The accelerating voltage for the electrons was 20 kV. The broadband dielectric spectroscopic data of samples were obtained using a broadband dielectric spectrometer model BDS-40, Novocontrol Technologies GmbH (Hundsangen, Germany), over a frequency range window of 10^−1^ Hz to 10^6^ Hz and at room temperature. The amplitude of the alternating current (A.C.) electric signal applied to the samples was 1 V.

## 3. Results and Discussion

Figure 1 presents the FTIR spectra of PMMI and PMMI-Li^+^, wherein it is possible to observe the characteristic absorption bands of PMMI. An absorption band attributed to the stretching carbonyl of ester groups at 1720 cm^−1^ was observed, while that attributed to the stretching of carbonyl of carboxylic acid was observed as a shoulder at 1650 cm^−1^. The strong absorption band at 1560 cm^−1^ corresponds to the vibration of methylene bonded to the ester group. Overall, there are no significant differences between the FTIR spectra of PMMI and PMMI-Li^+^, suggesting that the polymer would not interact with the Li^+^ ions. However, drastic differences in the Raman spectra of PMMI and PMMI-Li^+^ were observed (Figure 2). For instance, the band observed at 768 cm^−1^, which likely corresponds to the bending of methylene groups in the all-trans polymer backbone shifted to 827 cm^−1^ because of the presence of Li^+^ ions. This was probably due to the fact that strong interactions of PMMI with Li^+^ favor changes of the methylene groups to a mixed/gauche configuration [22]. Likewise, a band at 1420 cm^−1^ appears as a shoulder of the band observed at 1450 cm^−1^. The latter is associated with C=C stretching [23], so the appearance of the shoulder at lower frequency could indicate that the Li^+^ ions interact with these moieties of the polymer backbone. The presence of a new band at 1598 cm^−1^, associated with C=O stretching of carboxylate functions, also suggests that the Li^+^ ions interact directly with carboxylic acid moieties.

Figure 3a presents the real part (σ′) of the complex electrical conductivity (σ*) as a function of frequency of the electric field in the range between 10^−1^ and 10^6^ S cm^−1^. It was observed that the electrical conductivity of PMMI obeyed the Jonscher’s power law (Equation (1)).
(1)$$\sigma^{\prime }\left(\omega \right)=\sigma_{0}+A\omega^{s}$$
where σ_0_ is the D.C.-conductivity of the sample; $A\omega^{s}$ corresponds to the pure dispersive component of the A.C.-conductivity, characterized by presenting a power law in terms of angular frequency ω and exponent s (0 < *s* ≤ 1). The latter represents the degree of interaction between mobile charges and the molecular environment around them, and *A* is a constant associated with the polarizability strength [24]. The parameters of the fit curves according to the Jonscher’s power law of PMMI and PMMI-Li^+^ are shown in Table 1. By simple inspection, it is possible to infer that the presence of Li^+^ ions favors the charge transport process, evidenced by the increase of σ_0_ from 6.4∙10^−15^ to 4.8∙10^−11^ Scm^−1^. In addition, the change exhibited by *A* indicates that the Li^+^ ions likely favor the increase of the polarizability strength. Besides, the decrease of the exponent s could be associated with the influence of the Li^+^ ions in the system. It is important to mention that *s* exponent values exhibited by ionic and glassy semi-conductors are in the range of 1.0–0.6 [25].

A relaxation process at 10^2^ Hz was observed in spite of the fact that σ′ curve of PMMI-Li^+^ obeys the Jonsher’s power law. Figure 3b presents the permittivity as a function of the electric field frequency in the range between 10^−1^ and 10^6^ Hz. It is possible to observe that ε′ of PMMI is independent of the frequency, indicating that the pure ohmic conduction phenomenon takes place [26]. This, together with the fact that the dissipation factor was 0.02 (ν = 10^−1^ Hz), indicates that PMMI is a dielectric polymer [27,28]. Conversely, negative correlation of the frequency of ε′ was observed for the PMMI-Li^+^ sample, which could be attributed to multiple reasons, i.e., microscopic fluctuations of molecular dipoles; propagation of mobile charge carriers, such as translational diffusion of electrons, holes or ions; and the separation of charges at interfaces, favoring the occurrence of an additional polarization. The latter is probable related to the polarization that takes place between the sample and the electrode favored by the presence of Li^+^ ions (ν = 10^−1^ Hz). The relaxation process occurring ca. 10^2^ Hz, is also clearly seen in Figure 3c,d, and probably attributed to the dipole fluctuation favored by the interaction between Li^+^ ions and functional groups such as carboxylic acid and carboxlylate. Figure 3c also shows another relaxation in the high frequency domain (≈10^5^ Hz) notoriously seen in the PMMI sample. According to other authors, this broad relaxation (*β*-relaxation) could be attributed to the motions of ester groups directly attached to the polymer main [9].

Figure 4a,b presents the X-ray diffraction analysis and Raman spectra of the FGO, respectively. It is interesting to mention that the peak registered at 2θ = 25.3° is associated with the (002) reflection plane. By using the Bragg’s equation, the interlayer distance of FGO was determined as 0.351 nm. The wide width of this peak indicates (FWHM = 4.41°) the low crystallinity associated with a few layer graphene materials [29,30]. Raman spectra also provide information on this graphene material, which presents a low intensity *D* band (1348 cm^−1^), associated with the defects in the graphene lattice, such as vacancies and the presence of oxygenated functional groups. The fact that the intensity of the *2D* band (2685 cm^−1^) was almost half of that of the *G* band (1570 cm^−1^) confirms the few-layer nature of this graphenic material and suggests the presence of a long-range conjugated-π system in this material [31,32]. Moreover, Figure 5a–d presents SEM images of the PMMI, PMMI-Li^+^ and their nanocomposites containing FGO. It can be observed that the PMMI and PMMI-Li^+^ present similar morphologies. The addition of FGO to these matrices generates changes in morphology, as seen in Figure 5c,d. In addition, PMMI/FGO and PMMI-Li^+^/FGO nanocomposites partially differ in the distribution of the filler in their matrices. Both composites PMMI/FGO and PMMI-Li^+^/FGO present zones where it is possible to observe filler aggregates (see circled zones). 

Figure 6a,c presents the real part (*σ′*) of the complex electrical conductivity and the dielectric permittivity (ε′) PMMI/FGO nanocomposites as functions of the frequency of the applied electric field in the range between 10^−1^ and 10^6^ Hz. It was observed that the electrical conductivity of PMMI was drastically increased by the addition of 10 wt% of filler (σ′ = 3.02 × 10^−5^ S∙cm^−1^, recorded at ν = 10^−1^ Hz). This indicates that the percolation threshold was achieved for a FGO content higher than 5 wt%. The frequency independent behavior of the electrical conductivity indicates that the pathways of the mobile charge carriers, such as electrons, are favored by the presence of the graphenic material [17]. Undoubtedly, in the PMMI nanocomposite with 10 wt% FGO content, the layers tend to form a percolation network, where the charge carriers are transported by a process such as electron hopping. This indicates that the addition of FGO allows one to increase the electrical conductivity of PMMI, demonstrating the conductive nature of FGO as filler. Composites containing FGO < 5 wt% present curves similar to that exhibited by PMMI, but slightly shifted to higher conductivity values. This indicates that the graphene flakes present a contribution to the charge transport, but the filler particles are isolated or detached from other particles; consequently, the pathways that allow the electron hopping are not completely formed. The drastic increase of the permittivity (almost two-fold) in samples with FGO content of 5 wt% and below indicates that the polarization of inner dielectric boundary takes place on a mesoscopic scale due to the presence of FGO dispersed along the polymer matrix. This effect is known as Maxwell–Wagner–Sillar (MWS) polarization and can be explained based on the interfacial polarization that occurs at an interface between materials with different dielectrics constants, such as FGO and PMMI. Moreover, the slight decrease in the dielectric permittivity with frequency suggests that ion accumulation at the FGO/PMMI interface starts to appear.

On the other hand, Figure 6b presents σ′ in the frequency range of 10^−1^ and 10^6^ of the electric field of the nanocomposites based on PMMI-Li^+^/FGO, where significant differences compared with σ′ curves of PMMI/FGO are evidenced. It was observed that the electrical conductivity changed by four orders of magnitude for FGO contents as low as 5 wt%. However, the electrical conductivity of the composite containing Li^+^ and 10 wt% of FGO (σ′ = 4.37 × 10^−6^ S∙cm^−1^, recorded at ν = 10^−1^ Hz) was one order of magnitude lower compared to that exhibited by PMMI/FGO (10 wt%). This suggests that the presence of ions affects the electron hopping phenomenon, acting as charge transport carriers. Figure 6d presents the dielectric permittivities of those composites based on PMMI-Li^+^/FGO containing up to 5 wt% of filler. It was observed that the decrease in ε′ as the frequency increased was more notorious than that observed in PMMI/FGO nanocomposites. This indicates that multiple processes of polarization are simultaneously taking place, i.e., MWS and electrode polarizations and the probable dipole fluctuations attributed to the interaction between Li^+^ ions, the carboxylic acid and carboxlylate groups of the polymer. Moreover, the presence of FGO seems to be masking the relaxation process observed and previously discussed in the PMMI-Li^+^ matrix.

In order to further analyze the nature of the relaxation processes, dielectric loss factor (*ε″*) and loss tangent (*tanδ*) curves of PMMI/FGO and PMMI-Li^+^/FGO composites are shown in Figure 7. 

Starting with the PMMI/FGO nanocomposites, in the high frequency domain, the presence of FGO does not seem to have an effect on the *β-*relaxation of PMMI, as shown in Figure 7a,c. This means that the presence of graphene layers does not restrict or favor the motions of 1,1-side-carboxylic acid and ester groups of PMMI, although some other authors have reported that graphene layers can interact and affect the relaxation processes of functional groups of polar polymers [33]. At lower frequencies, the dielectric loss factor curves and the loss tanδ manifest high values, which can be ascribed to the contributions of multiple processes, i.e., electrode and MWS polarizations. 

On the other hand, similarly to those observed in ε′ curves, significant increases of ε″ at low frequencies were exhibited by samples based on PMMI-Li^+^ /FGO nanocomposites, as seen in Figure 7b. This is mainly attributable to the influence of Li^+^ ions; however, the strong polarization at low frequency hinders a thorough interpretation on the nature of the contributions. By considering the loss tangent (*tanδ*) curves (Figure 7d), it is possible to clearly identify a relaxation process ca. 10^0^–10^1^ Hz, which is probably due to the influence of graphene boundaries on the dipole fluctuation between Li^+^ ions with carboxylic and carboxylate groups. The graphenic layers restrict the motions of the dipoles and shift the maxima of the relaxation to lower frequencies compared to the PMMI-Li+. Moreover, the drastic increases of the loss *tanδ*, compared with those of PMMI/FGO nanocomposites, also reveal that the Li^+^ ions could act as charge transport carriers at the mesoscopic scale, producing dissipation of the electric field. 

Table 2 presents the parameters of Jonscher’s power law Equation (1) for composites with FGO content of 5 wt% or lower. It is possible to observe that these parameters are strongly influenced by the presence of Li^+^ ions. The significant differences between the two type of composites indicate that the Li^+^ ions play a role in the polarizability strength and the interaction of the mobile charge carriers and molecular environment around them. Furthermore, the change of the *s* exponent in samples containing Li^+^ suggests that the ionic character of a sample plays a role in the charge transport. This is also evidenced by the significant increase of the *tanδ* of those composites containing Li^+^. Regarding the maxima of *tanδ* (*tanδ_max_*) observed in nanocomposites based on PMMI-Li^+^/FGO, it is possible to infer that electrical relaxations take place. When considering a successful charge transport, the hopping of charge carriers to a new site is followed by the polarization cloud. If the latter does not occur, it is highly probable that the charge carrier will jump back [26]. In the PMMI-Li+/FGO nanocomposites, it can be speculated that the Li^+^ ions act as charge carriers at the mesoscopic scale and induce polarization of clouds. This would hinder the charge transport by electron hopping. Moreover, it is interesting to mention that the composite based on PMMI/FGO (3.0 wt%) presents *tanδ_max_* almost as low as that of pristine PMMI. This is probably attributable to the high affinity of PMMI with the filler because of the nature of the functional groups present in FGO. Furthermore, the increase of the ε′ due to the addition FGO to PMMI is in agreement with that sub-percolative approach using electrically conductive fillers for increasing the dielectric properties of dielectric polymers [18]. Finally, these interesting results can be relevant for the understanding and development of new sustainable and lightweight electrode and capacitors based on polymer composites.

## 4. Conclusions

The presence of FGO in nanocomposites based on PMMI and PMMI-Li^+^ imparted enhanced electrical conductivity, especially for those nanocomposites containing 10 wt% of FGO as filler. The dielectric permittivity of nanocomposites containing 5.0 wt% of FGO increased twice without exhibiting a significant increase of the loss tangent. Conversely, in composites containing Li^+^ ions, the permittivity was drastically increased, concomitant with the increase of the loss tangent. The presence of a relaxation process (ν ca. 10^1^ Hz) observed in *tanδ* suggests that the Li^+^ ions act as charge carriers at the mesoscopic scale and induce the polarization of clouds. This fact probably hinders the electron hopping due to the presence of Li^+^ ions in the pathways of electric percolation networks of FGO.

## Figures and Tables

**Figure 1 polymers-12-02673-f001:**
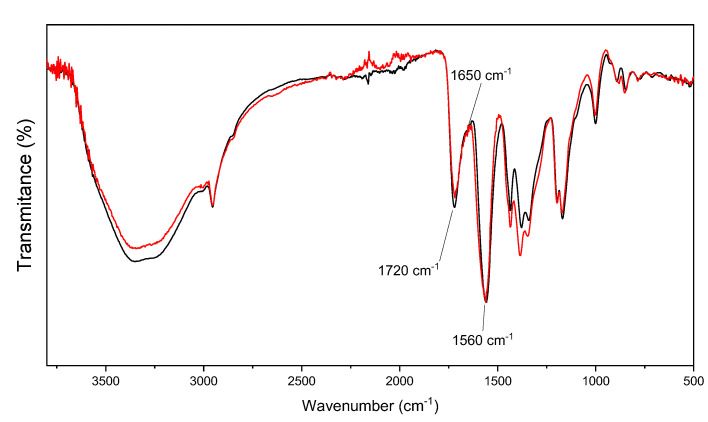
FTIR spectrum of PMMI (black line) and PMMI-Li^+^ (red line).

**Figure 2 polymers-12-02673-f002:**
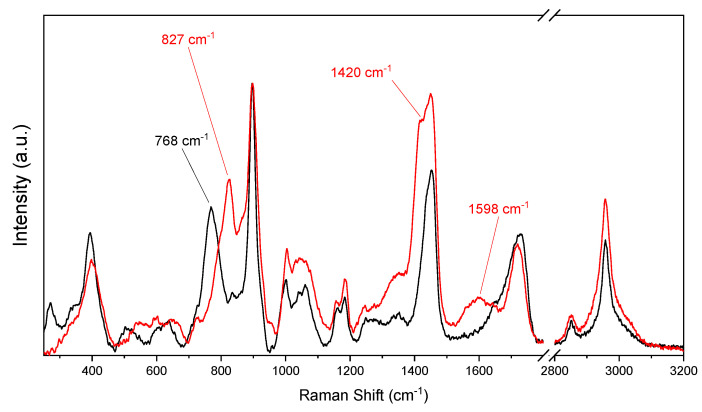
Raman spectrum of PMMI (black line) and PMMI-Li^+^ ions (red line).

**Figure 3 polymers-12-02673-f003:**
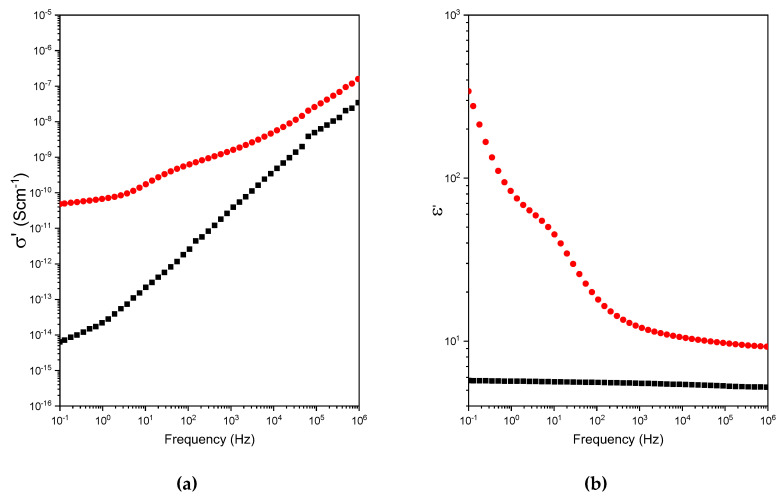
(**a**) Electrical conductivity (*σ′*), (**b**) dielectric permittivity (*ε′*), (**c**) dielectric loss (*ε″*) and (**d**) loss tangent (*tanδ*) as functions of the frequency for PMMI (black symbols) and PMMI-Li^+^ (red symbols).

**Figure 4 polymers-12-02673-f004:**
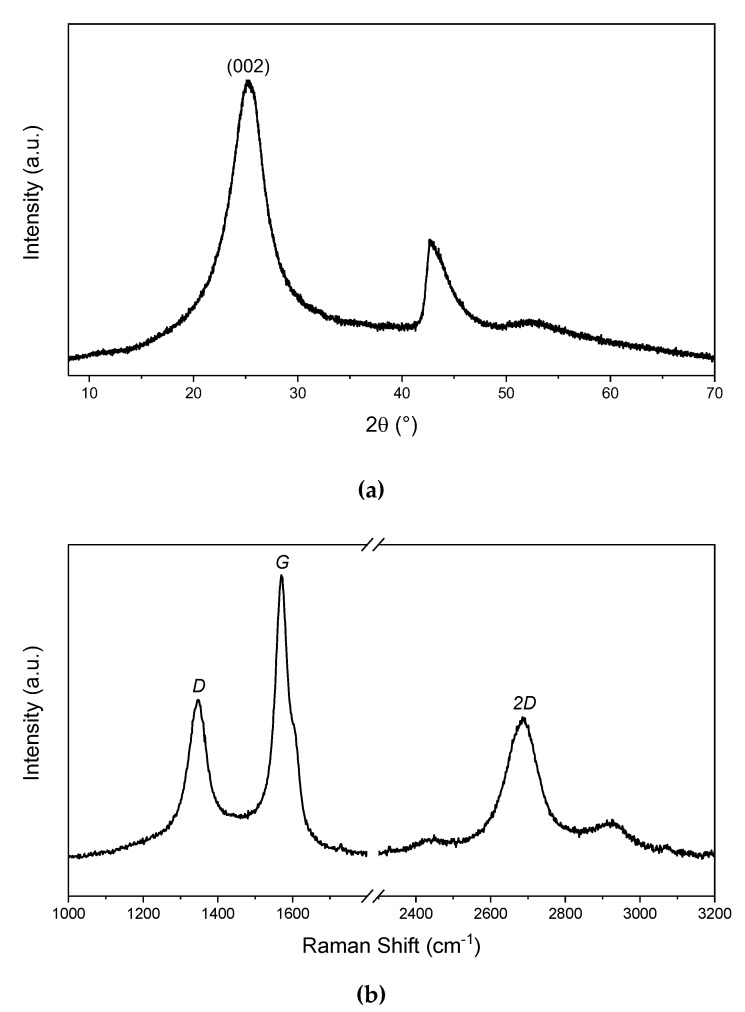
(**a**) X-ray diffraction pattern and (**b**) Raman spectrum of FGO.

**Figure 5 polymers-12-02673-f005:**
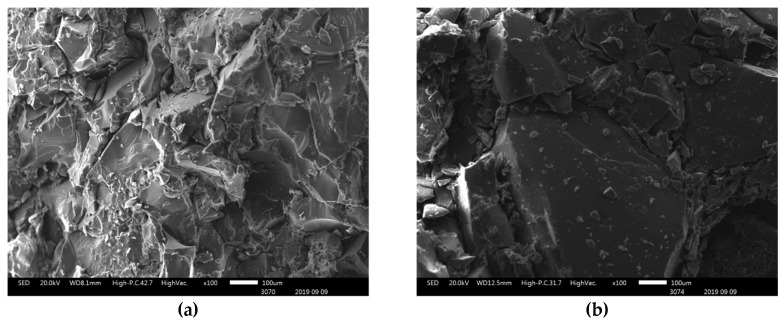
SEM images of (**a**) PMMI, (**b**) PMMI-Li^+^, (**c**) PMMI/FGO and (**d**) PMMI-Li^+^/FGO. Circled zones show filler aggregates.

**Figure 6 polymers-12-02673-f006:**
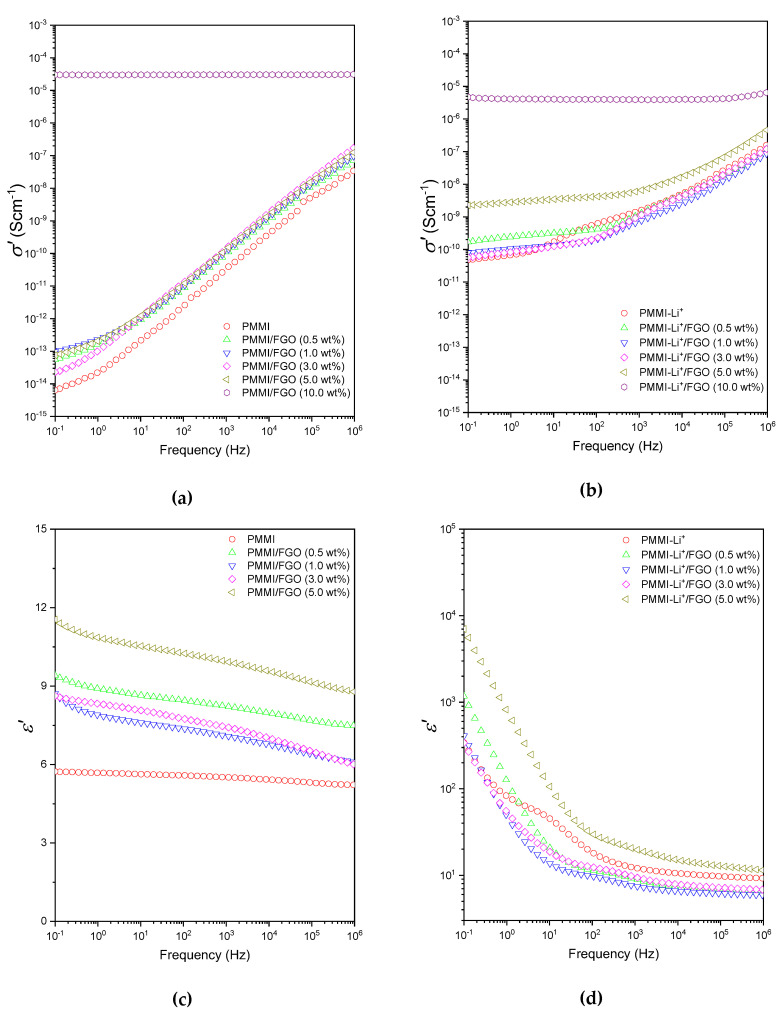
(**a**) Electrical conductivity (σ′) and (**c**) dielectric permittivity (ε′) as functions of the frequency for PMMI/FGO. (**b**) Electrical conductivity (σ′) and (**d**) dielectric permittivity ε′ as functions of the frequency for PMMI-Li^+^/FGO.

**Figure 7 polymers-12-02673-f007:**
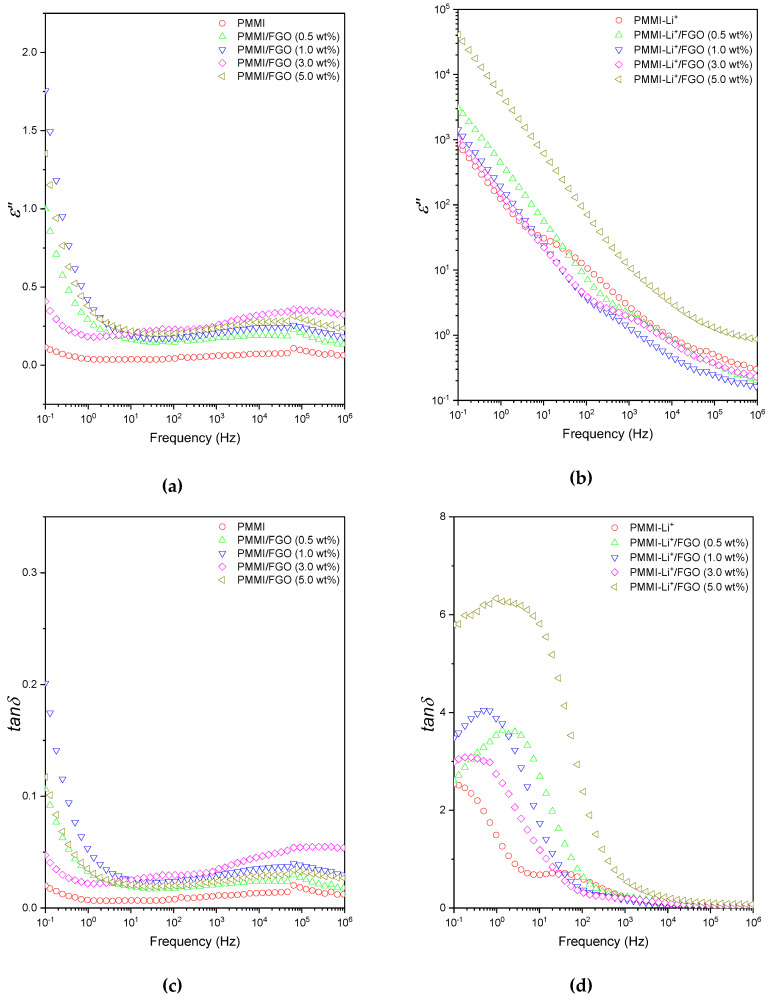
Dielectric loss factor as function of frequency for (**a**) PMMI/FGO and (**b**) PMMI-Li^+^/FGO nanocomposites; loss tangent (*tan δ*) as a function of frequency for (**c**) PMMI/FGO and (**d**) PMMI-Li^+^/FGO nanocomposites.

**Table 1 polymers-12-02673-t001:** Jonscher’s power law parameters of fit *σ′* curves of PMMI and PMMI-Li^+^.

Sample	σ_0_	*A*	*s*	R^2^
PMMI	6.4∙10^−15^ S cm^−1^	9.10∙10^−15^	0.971	0.993
PMMI-Li^+^	4.8∙10^−11^ S cm^−1^	7.65∙10^−13^	0.784	0.999

**Table 2 polymers-12-02673-t002:** Jonscher’s parameters, ε′, *tanδ* and *tanδ_max_* of nanocomposites.

Sample	σ_0_ (S cm^−1^)	*A*	*S*	R^2^	ε′	*tanδ*	*tanδ_max_*
PMMI	6.4∙10^−15^	9.10∙10^−15^	0.971	0.993	5.733	2.0∙10^−2^	2.0∙10^−2^
PMMI-Li^+^	4.8∙10^−11^	7.65∙10^−13^	0.784	0.999	341.0	2.53∙10^0^	2.53∙10^0^
PMMI/FGO (0.5 wt%)	5.7∙10^−14^	3.45∙10^−14^	0.933	0.995	9.407	1.06∙10^−1^	1.06∙10^−1^
PMMI/FGO (1.0 wt%)	5.2∙10^−14^	3.39∙10^−14^	0.954	0.997	8.719	2.02∙10^−1^	2.02∙10^−1^
PMMI/FGO (3.0 wt%)	2.3∙10^−14^	3.05∙10^−14^	0.997	0.999	8.623	4.73∙10^−2^	4.73∙10^−2^
PMMI/FGO (5.0 wt%)	7.6∙10^−14^	3.23∙10^−14^	0.973	0.999	11.55	1.17∙10^−1^	1.17∙10^−1^
PMMI-Li^+^/FGO (0.5 wt%)	1.7∙10^−10^	7.59∙10^−13^	0.759	0.999	1175	2.60∙10^0^	3.63∙10^0^
PMMI-Li^+^/FGO (1.0 wt%)	7.8∙10^−11^	2.24∙10^−13^	0.824	0.999	409.2	3.48∙10^0^	4.04∙10^0^
PMMI-Li^+^/FGO (3.0 wt%)	5.6∙10^−11^	1.52∙10^−12^	0.702	0.997	339.1	2.99∙10^0^	3.08∙10^0^
PMMI-Li^+^/FGO (5.0 wt%)	2.3∙10^−9^	4.95∙10^−12^	0.728	0.993	7014	5.80∙10^0^	6.33∙10^0^

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
