# Peer review of "Electrical Properties of Poly(Monomethyl Itaconate)/Few-Layer Functionalized Graphene Oxide/Lithium Ion Nanocomposites"

_polymers, 2020, doi:10.3390/polym12112673_

Round 1
Reviewer 1 Report
The article presents electrical studies concern the polymer composites based on a graphenic material and poly(monomethyl itaconate) in the presence and absence of Li+ ions synthetized by the authors. The manuscript is well-written and the experimental procedures and results presents well discussed. It is recommended the acceptance of the paper in Polymers journal after minor corrections:
(1) Please indicate in references topic the cited Patent Cooperation Treaty (PCT);
(2) Page 2, line 90: "Finally, the solid was dried at 40 °C under vacuum until reaching constant weight.", please inform the vacuum pressure and equipment.
(3) Please insert the units of the non-dimension quantities in table 2. In addition, keep numeric formatting or in scientific or decimal notation for the last two columns.
Reviewer 2 Report
Recommendation: Reconsider after major revision
Comments:
This work compares the electrical conductivity properties of poly(monomethyl itaconate)/few-layer functionalized graphene oxide nanocomposites in the presence and absence of Li+. This has a certain significance for the development of new sustainable lightweight electrodes and capacitors. However, some important experiments and data are not provided in the result section. So, the reviewer thinks it is recommended to reconsider after major revisions as specified below.
- The background and the relevant references in the introduction section are not sufficient, please add more information about your research.
- Please add the structural and thermal performance characterization of PMMI and PMMI-Li+, such as 1H NMR, DSC, TGA, please add.
- In Page 8, from figure 6(a)Electrical conductivity、(b) Electrical conductivity (σ’), there is no obvious mutation from the addition of 0-5wt% FGO, but there is an obvious mutation between 5-10wt%. There are no data points in the interval (5wt%-10wt%), so the reviewer insists the conclusion drawn from this is Incomplete. It is recommended to add at least two points in this interval.
- The reference format needs to be unified.
Round 2
Reviewer 2 Report
The authors have revised their manuscript. Thanks!